

# Silver nanoparticles can be sampled by ultrafiltration probe but elution into & recovery from plasma and Dulbecco's Phosphate Buffered Saline differs *in vitro*

Marije Risselada[1], Robyn R. McCain[2], Miriam G. Bates[1] and Makensie L. Anderson[1]

[1] Department of Veterinary Clinical Sciences, College of Veterinary Medicine, Purdue University, West Lafayette, Indiana, United States
[2] College of Veterinary Medicine, Purdue University, Center for Clinical Translational Research, West-Lafayette, Indiana, United States

## ABSTRACT

Locally sustained release antimicrobials such as silver nanoparticles (AgNPs) might prove useful in combating local infections. Their elution has been investigated in Phosphate Buffered Saline (PBS) including from poloxamer 407 (P407). No information exists on possible interactions with proteins, nor have ultrafiltration (UF) probes been evaluated to measure Ag. These investigations would provide helpful data prior to investigating the sustained release after subcutaneous implantation *in vivo* over time. We compared (1) the influence of elution fluid on AgNP elution, and (2) UF probe and direct sampling *in vitro*. Six AgNP-P407 specimens in a dialysis tube were placed in Dulbecco's PPBS (DPBS) ($n = 3$) or canine plasma ($n = 3$) for 96 h on a stirred hot plate (37 °C and 600 rpm) and sampled 20 times. A 0.001 mg/mL AgNP solution was prepared in DPBS or plasma. Six pipette and UF probe samples were taken of each. Inductively coupled plasma mass spectrometry was used to analyze Ag. Stock plasma contained Ag. At 96 h, 5/6 dialysis tubes had not fully released AgNP. One peak in hourly Ag increase was present in DPBS (10–13 h), and two peaks in plasma (6–8 and 10–13 h). The hourly Ag increase in plasma decreased earlier than in DPBS. Ultrafiltration probe sampling was possible in both DPBS and plasma and resulted in higher Ag concentrations but with more variation. While *in vitro* use of DPBS might be more cost effective, plasma should be considered due to difference in elution and recovery. Ultrafiltration probes can be used to sample Ag, but results will have a greater degree of variation, and multiple samples and increased time points should be considered.

# INTRODUCTION

With the increase in antimicrobial resistance in veterinary medicine (*Perrenten et al., 2010*; *Cohn & Middleton, 2010*), novel strategies and antimicrobials to combat infections are (re)

Corresponding author
Marije Risselada,
mrissela@purdue.edu

gaining favor[1]. One strategy might be to deliver a high local dose of antibiotics (*Ham et al., 2008*; *Peterson et al., 2021*; *Lee, Frederick & Cross, 2019*; *Reed et al., 2016*; *Smith et al., 2023*). Another might be to explore the use of non-antibiotic antimicrobials, such as silver (Ag), especially silver nanoparticles (AgNP). They have received considerable interest for their use against micro-organisms (*Li et al., 2010*; *Senthilkumar et al., 2018*) and in wound care (*Augustine et al., 2018*). Several sustained release carriers have been described for drug delivery, including delivery of AgNP, such as calcium sulfate beads, poloxamer 407 (P407) and a gelatin sponge (*Peterson & Risselada, 2022*). Poloxamer 407 is a versatile reverse gelatinating polymer that is safe for implantable use (*Mathews et al., 2009*; *Risselada et al., 2017*), and has found use for delivery of antifungals (*Risselada et al., 2017*), chemotherapeutics (*Mathews et al., 2009*; *Risselada et al., 2020*), and antibiotics (*Risselada, Spies & Kim, 2024*). It serves as a local depot from which the drugs are slowly absorbed into the surrounding tissues. Poloxamer 407 also might have inherent antimicrobial properties of its own (*Bates et al., 2024*). Prior silver elution studies, including from P407 have been performed into phosphate-buffered saline (PBS) from poloxamer 407 (*Peterson & Risselada, 2022*), into deionized water baths (from silver oxide films) (*Goderecci et al., 2017*), or aqueous samples with varying NaCl concentrations (from solid coated specimens) (*Kent & Vikesland, 2012*). Plasma might be a closer *in vitro* fluid approximation to an *in vivo* environment than PBS and could mimic protein interactions that might be encountered. The elution into human plasma from an antibiotic containing polymethyl methacrylate (PMMA) construct as well as the effect plasma on PMMA properties was reported (*Schmidt-Malan et al., 2019*). However, plasma has not been evaluated for suitability of Ag release studies, and whether the benefit could outweigh the increased cost of commercial canine plasma (~$2,400/500 ml; Innovative research, Novi, MI) over PBS (~$30/500 ml; Gibco, Thermo Fisher Scientific, Waltham, MA, USA).

The ultimate goal of sustained release compounds assessed by elution studies is their *in vivo* application. Ultrafiltration (UF) probes have been used successfully in dogs (*Bidgood & Papich, 2002*; *Papich, Davis & Floerchinger, 2010*; *Maaland, Guardabassi & Papich, 2014*; *Messenger, Wofford & Papich, 2016*) and pigs (*Messenger, Papich & Blikschlager, 2012*). The most used UF probe consists of three 6-cm long loops comprising 12 cm of semi permeable membrane per loop for a total of 36 cm of membrane (RUF 3–12 Reinforced Ultrafiltration Probe; BASi Instruments, West-Lafayette, IN, USA). The semi permeable membrane allows molecules of up to 30 kDa to pass through by the negative pressure generated in a vacutainer blood tube. These UF probes are used to sample compounds present in the tissue by obtaining interstitial fluid (ISF), without the need for more invasive sampling methods, such as tissue biopsies (*Maxwell et al., 2020*).

Establishing the feasibility of sampling Ag with UF probes *in vitro* would be preferable prior to their use to investigate the *in vivo* pharmacokinetic profile of Ag containing local delivery products. However, the possibility of sampling AgNP *via* UF probe has not been assessed, nor has a comparison of canine UF probes and direct sampling been reported for AgNP. Similarly, no data exists on whether presence of proteins could interfere with UF probe sampling of Ag.

**Figure 1** **Diagram depicting the formulation of AgNP-P407: 2.5 ml of AgNP (shaded) was added to 5.0 ml of poloxamer 407 (P407, gray) to create the solution.** The gray shaded color will be used throughout the diagrams.

**Figure 2** **Creation of the specimens.** A 10 cm length of dialysis tube was precut, and one end folded over (A), and secured in place by two hemoclips (B). The solution was then injected into the tube (C), the free end folded over (D) and secured (E) with two hemoclips.

The intent of the study was to address several gaps in one *in vitro* study prior to performing *in vivo* sampling studies. Specifically, we aimed to (1) determine and compare the influence of elution fluid on rate, pattern, and completeness of AgNP elution from P407 *in vitro*, and (2) compare UF probe sampling with direct samples. We hypothesized that (1) elution of AgNP into Dulbecco's PBS (DPBS) will be similar to elution in canine plasma, and that (2) UF probe sampling will be constant over time and yield similar results as direct sampling.

## MATERIALS AND METHODS

An observational elution study and repeat sampling study were performed separately. Canine Ultrafiltration Probes with three loops of 12 cm membrane (RUF 3-12) were assembled as per manufacturer instructions (*BASi*).

### Elution

A commercial solution of 0.02 mg/ml 10 nm silver nanoparticles (AgNP) in aqueous buffer with sodium citrate as stabilizer (Sigma Aldrich, St. Louis, MO, USA) was used for the study. Six elution specimens were prepared (Fig. 1). Each contained 2.5 ml of the 0.02 mg/ml 10 nm AgNP stock solution mixed with 5.0 ml 30% poloxamer 407 (Pluronic® F-127; Sigma-Aldrich, Burlington, MA, USA) (1:2 ratio with a total Ag content of 0.05 mg/specimen). The 1:2 ratio was chosen as the ratio with highest AgNP content that would still fully gel at room temperature, based on prior observations of a 1:1, a 1:2 and a 1:3 ratio of AgNP to poloxamer 407 after observations that a 1:1 ratio did not fully gel at room temperature, but only at 37 °C (*Bates et al., 2024*). The 1:2 and 1:3 ratios fully gelled up at room temperature (personal observation).

The specimens were prepared in individual 12 ml syringes <2 h before use and stored refrigerated and shielded from light. Specimens were created (Fig. 2) using a 10 cm long strip of 1 inch dialysis tubing with 12–14 kDa pores (Carolina Biological Supply Co, Burlington, NC, USA) (*Risselada et al., 2016*). The distal free end was folded up length wise and secured in folded position using two large surgical stainless steel hemoclips (Hemoclips®; Teleflex, Morrisville, NC, USA) in opposite direction (*Peterson & Risselada,*

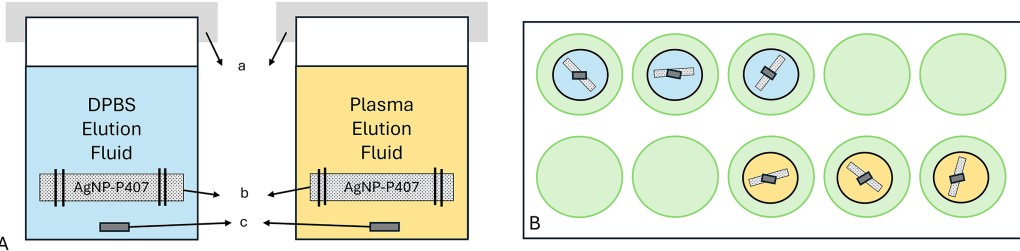

**Figure 3 Elution setup with the individual crystallization dishes (A), and their location on the hot plate (B).** Each dish was covered by aluminum foil (a), tightly closed around the rim. The specimen (b) was placed in the prewarmed elution fluid, with the spinning magnet (c) already in place.

*2022*). The 1:2 AgNP-P407 mix was then placed in the tube, and the proximal free end of the tube folded and closed in similar fashion. All specimens were created in one batch and then placed in the prewarmed DPBS or canine plasma within 5 min after assembly. Individual 150 ml crystallization dishes (Synthware glass, Pleasant Prairie, WI, USA) with 100 ml fluid were prewarmed for 2 h prior to starting the elution study. Three dishes had Dulbecco's phosphate-buffered saline without Calcium chloride or magnesium chloride added (DPBS, Gibco, Thermo Fisher Scientific, Waltham, MA, USA) and three had canine plasma with dipotassium EDTA (K2-EDTA) (IGCNPLAK2E 500 ml Canine Plasma lot 41,273, Innovative research, Novi, MI). A 10 multi-position hotplate with magnetic stir bars (Fig. 3) was used (RT 10, IKA Magnetic Stirrers, Wilmington, NC) with settings at 37 °C and 600 rpm throughout the experiment. Room temperature was set at 68 °F. The t = 0 sample was taken immediately after all specimens were submerged in the same order of assembly and placement. Twenty samples (0.15 ml each) were taken over 96 h with a decrease in frequency (0, 1, 2, 3, 4, 5, 6, 8, 10, 13, 17, 22, 27, 34, 42, 48, 58, 66, 72, 96 h) with the sampling order consistent throughout. At 96 h, an additional sample (by needle aspiration through the tube) of the fluid contained within the dialysis tube was taken.

### Ultrafiltration probe sampling

Two specimens were prepared immediately prior to sampling in a 50 ml conical centrifuge tube (Thermo Fisher Scientific, Waltham, MA, USA) using graduated pipettes (Thermo Fisher Scientific, Waltham, MA, USA) and a 3 ml syringe (Covidien, Mansfield, MA, USA). Each specimen contained 0.03 mg of AgNP (1.5 ml of the commercial 0.02 mg AgNP stock solution) total in 28.5 ml of either DPBS (*n* = 1) or canine plasma (*n* = 1) for a targeted fluid concentration of 0.001 mg/ml of AgNP. Ultrafiltration probes were tested for patency and sampling using DPBS prior to use for the study.

Full submerging of the three 6 cm long loops of the UF probe near the lowest point of the tube was ensured, and positioning was checked after each manipulation. Vacutainers without additives (Vacuette Blood collection tube, 3.0 ml, no additive, Greiner Bio-One; Sigma-Aldrich, St Louis, MO, USA) were used to collect the probe samples and an additional 10 ml of air was removed to increase negative pressure in each vacutainer to increase sampling speed. The initial sample of the study specimen was discarded to avoid risk of dilution by plain DPBS. Repeat samples using both a UF probe and direct sampling

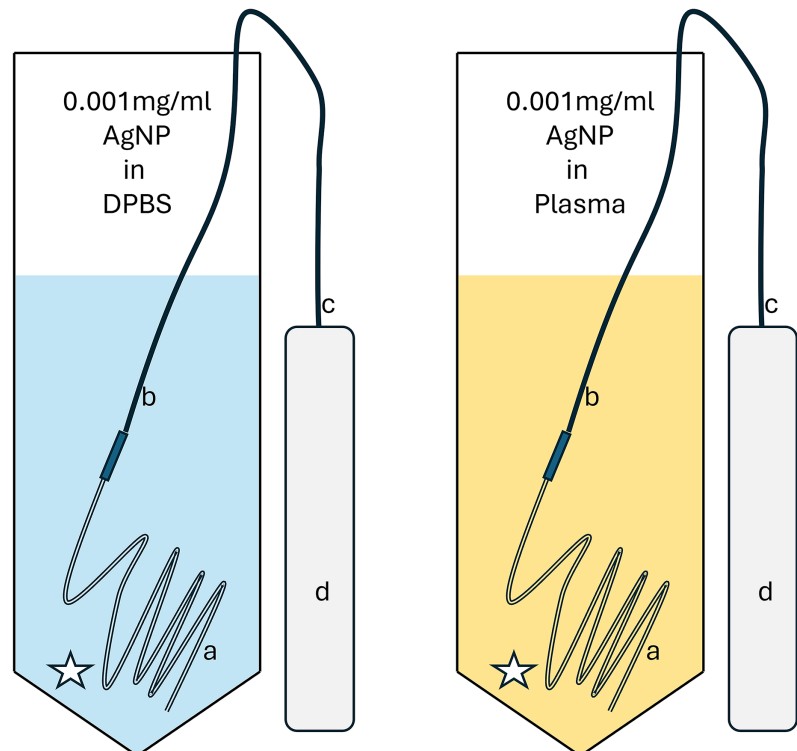

**Figure 4** **Repeat sampling diagram for both 0.001 mg/ml AgNP in DPBS (blue) and plasma (yellow).** A "star" depicts the location for the pipette sampling. A 3-12 UF probe (three loops of membrane, each 12 cm in length) is depicted with the membrane (a), the connection to the tube (b), the tube itself connecting *via* a needle (c) to a vacuum tube (d).

using a pipette from the same area as the membrane of the UF probe were taken at 0, 5, 10, 15, 30 and 60 min from the plasma and DPBS specimen (Fig. 4).

Samples of commercial stock AgNP solution, DPBS and canine plasma were taken prior to the start of the study. All samples were stored at −78 °C until batch analysis.

## Sample and data analysis

The quantity of Ag in each sample was determined *via* inductively coupled plasma mass spectrometry (ICP-MS) as previously described (*Bates et al., 2024*). DPBS samples were diluted in 2% $HNO_3$ and plasma samples were digested over night at 70 °C in the 1:1 mixture of 70% $HNO_3$ and $H_2O_2$ and analyzed using ICP-MS (Perkin Elmer NexION 300D) to determine the concentration of silver within each sample. The short-term precision was less than 3% relative standard deviation (RSD), and the long-term stability was <4% relative standard deviation over 4 h. Isotope-ratio precision was less than 0.08% relative standard deviation. The Ag detection limit was 0.001 ng/ml, and quantification limit at 0.002 ng/ml, and all samples below this limit were recorded as 0 ng/ml. Silver concentrations were expressed in parts per billion (ppb), with 1 ppb = 1 ng/ml. Hourly increase in Ag was calculated as the difference between the measured values at two subsequent time points divided by the hours between time points (expressed as ppb/hr). The data of the three elution specimens will be expressed graphically as mean ± SD. The

**Table 1 Silver (Ag) at 96 h expressed as parts per billion (ppb) for the remaining elution fluid outside of the dialysis tube and the remaining specimen contained within the dialysis tube.**

| Ag in DPBS | | Ag in plasma | |
|---|---|---|---|
| Fluid | Specimen | Fluid | Specimen |
| 2137.97* | 203.32* | 25 | 32.34 |
| 293.18 | 512.65 | 20.02 | 68.29 |
| 248.75 | 528.26 | 23.36 | 58.90 |

Note:
Each tube contained 6,660 ppb Ag at the start of the experiment. An asterisk (*) denotes the specimen that was not fully filled and the surrounding elution fluid. Silver at 96 h was analyzed once for six specimens (three each in DPBS and plasma) and elution fluid.

values of the repeat sampling experiment will be reported as individual results and a mean ± SD (RSD) for both specimens in an observational manner without further statistical analysis. The RSD was calculated by dividing the SD by the mean and will be expressed as a %.

## RESULTS

Silver concentrations for stock solutions used in this study were: DPBS stock 0.19 ppb Ag; plasma stock 4.83 ppb Ag and the commercial 0.02 mg/ml AgNP solution contained 24,940 ppb Ag. Assembled UF probes initially did not reliably yield appropriate negative suction to obtain a sample, and probes were not re-usable for repeat experiments. Additional negative pressure applied to the vacutainers together with sealing connecting points with glue (3 g Tube; The Gorilla Glue Company, Cincinnati, OH, USA) allowed single experiment sampling.

### Elution

No leakage of any dialysis tubes was observed under gentle pressure immediately after assembly. Five out of six dialysis tubes were fully filled at 96 h, the sixth was not fully filled when retrieved. The five fully filled specimens still contained more Ag than the surrounding fluid at 96 h (Table 1), and release of Ag was not complete at 96 h. A burst release of Ag was seen both into DPBS and plasma in the first 13 h (DPBS, light grey) and 8 h (plasma, dark grey), with the baseline amount in plasma higher than in DPBS (Fig. 5). The increase of Ag measured in plasma decreased earlier, between 8–60 h (dark grey, Fig. 5), and the measured Ag decreased thereafter. The hourly increase of Ag in DPBS (light grey) tapered between 27–84 h but continued throughout the study (Fig. 6). The highest hourly increase in Ag was between 10–13 h in both DPBS (light grey) and plasma (dark grey) (Fig. 6). A clear single peak in DPBS (20.66 ppb/hr) whereas two peaks were present for plasma: between 6–8 h (12.31 ppb/hr) and between 10–13 h (13.85 ppb/hr) (Fig. 5). The amount of Ag measured at 96 h was higher in the remaining AgNP:P407 mix contained within the tube than the surrounding fluid (both DPBS and plasma) (Fig. 7). The Ag concentration measured at 96 h in DPBS was higher (both specimen and fluid) than in plasma (Fig. 2). The mean ± SD total amount of Ag removed *via* sampling for the

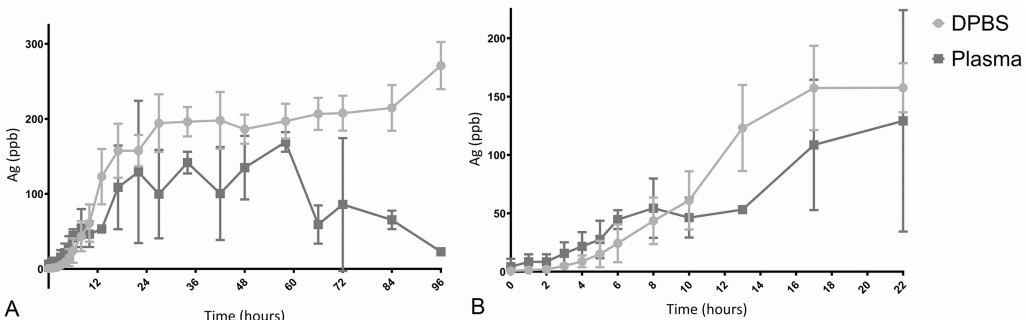

**Figure 5  Elution of Ag into DBPS or plasma.** (A) Elution of Ag into either DPBS (light grey) or plasma (dark grey), expressed as ppb Ag over 96 h. The elution of Ag into DBPS continued longer, leading to a higher fluid concentration, whereas the increase of Ag in plasma tapered between 24–60 h and Ag decreased after 60 h. (B) Elution of Ag into either DPBS or plasma, expressed as ppb Ag over the first 22 h of the study.

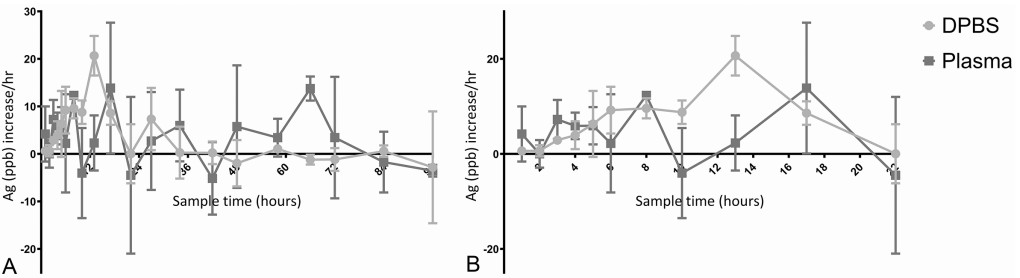

**Figure 6  Hourly increase of Ag into DPBS or plasma.** (A) The increase in Ag is expressed as Ag (ppb)/h measured over the timeframe prior to the sampling time. A burst elution pattern is evident for both elution fluids. (B) Hourly increase in Ag over the first 22 h. The measured values of Ag in plasma varied more for each time point and between time points as evidenced by the larger bars and the shape of the curve.

three DPBS fluid set ups was 330 ± 51 ng Ag and for the three plasma fluid set ups 198.1 ± 71 ng Ag.

## Ultrafiltration probe sampling

Anticipated Ag content based on the stock (24,940 ppb Ag (AgNP stock) diluted in 30 ml) was 1,247 ppb Ag of the diluted fluid. Ultrafiltration probes were able to collect Ag in both DPBS and plasma, with Ag content in fluid obtained *via* UF probe sampling higher than the corresponding pipette samples (Fig. 8). However, the UF probe-obtained samples had more variation and values differed between sampling times and decreased over time (Table 2). Samples obtained by both methods had a measured Ag lower than anticipated, with the underestimation greater in DPBS than plasma.

## DISCUSSION

Elution into DPBS yielded different results than elution into plasma, with a longer sustained initial hourly increase in fluid Ag and more variation in Ag measurements with a resultant less smooth curve. A burst release pattern was evident for both DBPS and plasma as elution fluid. The burst release corresponds with an earlier study where ~88% of the Ag

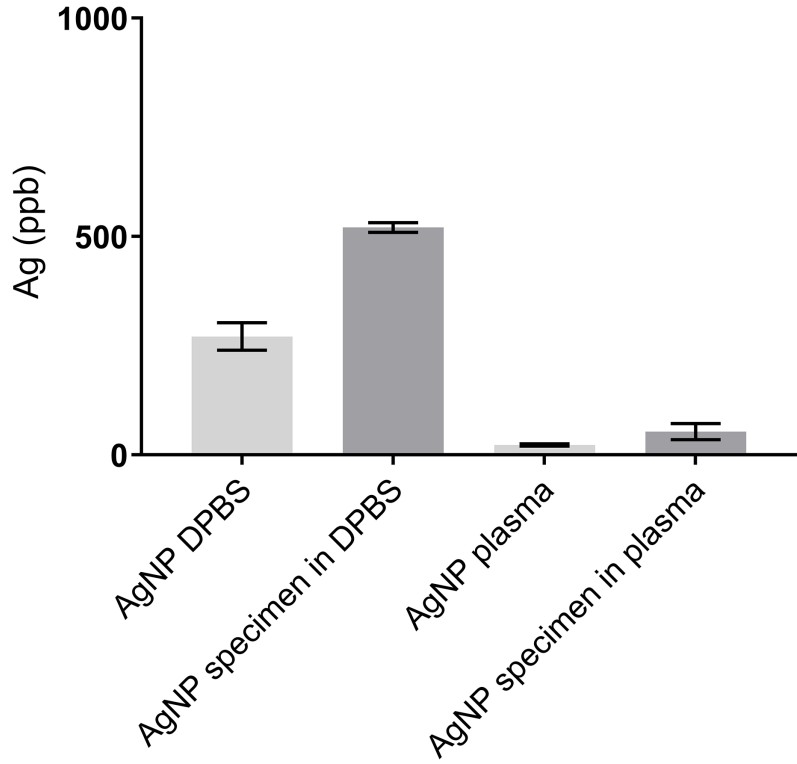

**Figure 7** **Ag amount (ppb) in elution fluid (DPBS or plasma) and remainder of the Ag:poloxamer 407 specimen at 96 h.** The mean ± SD of the measured Ag in the elution fluid is shown in light grey, and the Ag measured in the corresponding remaining specimen in darker grey. The plasma elution fluid is shown in dark gray. Measured Ag content in both the fluid and Ag:poloxamer 407 specimen remnant at 96 h were less in plasma than DPBS, with the specimens still higher in Ag content than the surrounding DPBS or plasma.                                              

was released from poloxamer 407 within the first 24 h (*Bates et al., 2024*). While the pore size of the dialysis tube should allow AgNPs to cross freely, the Ag concentration within the dialysis tube was still higher at 96 h than the surrounding fluid (both DPBS and plasma), indicating incomplete release of the specimen in both elution media. Silver could be measured in DPBS and plasma by UF probe sampling.

The higher initial values of Ag found in plasma might be explained by Ag being present in stock plasma while no Ag was found in the DPBS stock solution. The higher initial value in plasma was then followed by an increase in Ag content similar in shape to the DPBS curve. However, the presence of Ag, and the initial starting value, would mask the initial release of Ag at the time when the release will be highest and is a limitation of using plasma as the elution fluid. We chose to report the measured values as-is instead of detracting the initial Ag measured obtained from stock plasma, as we only had a single measurement for the stock plasma rather than repeat samples and thus felt that correcting could inadvertently introduce error as well.

Plasma with EDTA was chosen as it was the cheapest commercial option available. Ethylenediaminetetraacetic acid (EDTA) chelates silver (*Fulgenzi & Ferrero, 2019*), and its presence in plasma theoretically could help extract Ag from the specimen contained within

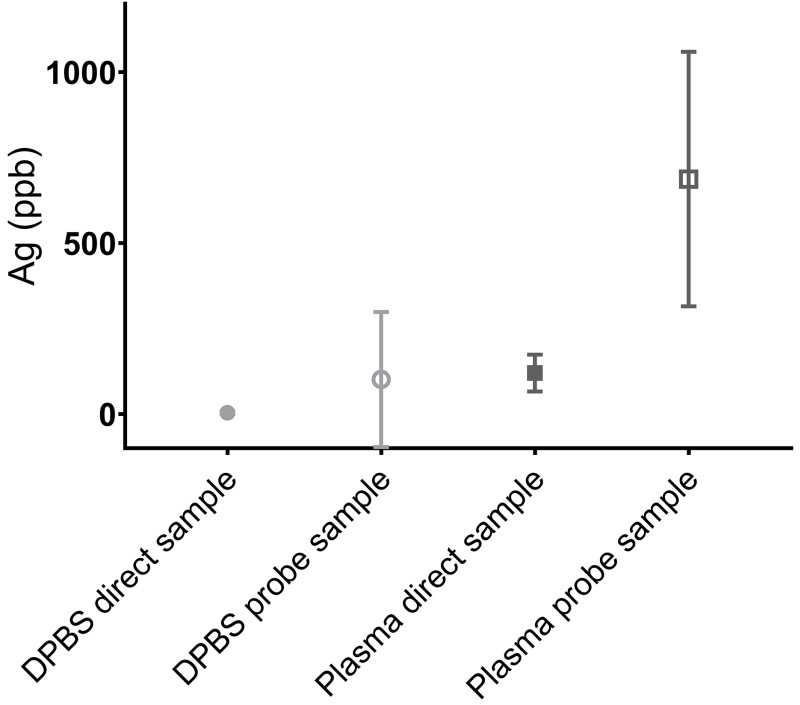

**Figure 8 Pipette and ultrafiltration (UF) probe sampling of 1,247 ppb Ag in DPBS (light grey) and plasma (dark grey) solution.** Pipette sampling is represented by a solid sphere (DPBS) or square (plasma) symbol, while the corresponding UF sampling data is shown by a solid sphere or square. Both methods underestimated the Ag content in both mediums. The samples obtained *via* UF probe had a wider variation in measured Ag. The dotted line indicates 1,247 ppb Ag.

**Table 2 Repeat pipette and ultrafiltration (UF) probe sampling of a planned solution of 1,247 ppb Ag in DPBS and plasma solution (one specimen each).**

| Repeat sampling of Ag (ppb) | AgNP in DPBS specimen ($n = 1$) | | AgNP in plasma specimen ($n = 1$) | |
| --- | --- | --- | --- | --- |
| Time (min) | UF probe | Pipette | UF probe | Pipette |
| 0 | 502.67 | 1.19 | 791.28 | 116.19 |
| 5 | 48.31 | 11.31 | 946.58 | 99.18 |
| 10 | 33.88 | 0.93 | 370.54 | 225.08 |
| 15 | 15.47 | 3.18 | 117.41 | 74.07 |
| 30 | 2.33 | 0.80 | 1111.30 | 116.02 |
| 60 | 3.95 | 6.93 | 785.83 | 89.27 |
| mean ± SD (RSD) | 101.10 ± 197.53 (195%) | 4.06 ± 4.25 (105%) | 687.2 ± 372.2 (54%) | 119.9 ± 53.9 (45%) |

**Note:**
Probes were allowed to sample for 4 min to obtain enough sample volume. Direct samples were taken using a pipettor at the time point. UF probe samples were taken by attaching a vacutainer to the probe for 4 min at the start time. DPBS = Dulbecco's Phosphate Buffered Saline, UF = ultrafiltration. The mean is provided with both standard deviation (SD) and relative standard deviation (RSD) for each sampling method over time from the same fluid.

the dialysis tube. However, chelation of Ag into stable complexes also might hinder analyses, although measuring of excreted EDTA-metal complexes before and during chelation therapy has been described (*Fulgenzi & Ferrero, 2019*; *Robotti et al., 2020*). In addition, $HNO_3$ digestion has been used to separate heavy metals from EDTA (https://patentimages.

storage.googleapis.com/5c/43/f6/ac08bfd470cea7/US20080038169A1.pdf). However, depending on the compound for which the elution is performed and its chelation abilities with EDTA plasma with a different anti-coagulation agent might be preferred.

We used a 1:2 ratio of AgNP:P407 in this study to maximize the Ag content while still maintaining the full gelatinating properties of the poloxamer. In a prior elution study, a 1:4 ratio of AgNP:P407 was used and the specimens fully gelatinated (*Peterson & Risselada, 2022*). However, specimens were observed to not fully gel at room temperature in a 1:1 ratio in a different study (*Bates et al., 2024*). The specimens did gel at room temperature in a 1:2 ratio based as the highest silver containing ratio at the time of the bactericidal study (personal observation) although this finding was not published as part of the bactericidal study, only the finding that the 1:1 ratio did not fully gel at room temperature (*Bates et al., 2024*). The specimen composition, as well as methodology (a single, larger amount of elution fluid with sampling as opposed to full exchange of smaller amounts of elution fluid) between this study and the prior study might account for the earlier tapering of elution and the flattening of the curve. The fluid exchange would increase the gradient and therefore drive migration of Ag across the dialysis tube membrane. The fluid quantity was chosen to allow a model with a larger amount of fluid without full exchange to allow continuous stirring, the ability to submerge a large specimen, and due to the cost of canine plasma (~$2,400 per 500 ml). Removing a sample at each time point (0.15 ml each for a total of 3 ml) could impact the Ag concentration by both removing fluid as well as Ag. The total amount of Ag removed over the entirety of the 96 h was 330 ng for the DPBS fluid set ups and 158 ng for the plasma set ups. Ideally the removed amount of Ag would have been added back in a corrected calculating using the currently present volume of the elution fluid, however, the fluid present at each time point was not measured, and using an approximation might lead to a flawed correction, and more error than not correcting.

The dialysis tube model was chosen to avoid the poloxamer from dissolving immediately upon placing the AgNP:P407 mix in the elution DPBS or plasma fluid. Prior studies with the same commercial AgNP and dialysis tubing (*Peterson & Risselada, 2022*) yielded appropriate migration of Ag through the membrane. Given that no plasma was placed within the tube, there were no concerns of Ag complexing with proteins, leading to inability to migrate across the membrane due to pore size (12–14 kDa).

While aluminum foil was wrapped tightly around the opening of the dishes, fluid loss, in addition to sampling loss due to warming of the fluid was possible. Aluminum foil was chosen as the initial intent was to perform all sampling with UF probes in addition to direct sampling and using snap-on or screw-on lids would have damaged the probes. The study was converted to direct sampling only at the last minute due to concerns of probe functioning and continued patency during the initial assembly, and fluid was already in the dishes being prewarmed. The evaporative fluid loss could account for the increase in Ag concentration of the fluid at the end of the study. No attempt was made to estimate fluid losses during the study, as measuring at each time point would have necessitated interrupting the elution and manipulating the specimen and might cause additional fluid loss. However, measuring the volume at 96 h could have been performed. Fluid loss during a continuous release elution model is a known limitation of the model chosen (*Risselada*

*et al., 2016*), however removing and replacing all fluid at each time point might artificially keep a larger gradient intact than a continuous model would. In addition, a continuous model might more closely resemble the *in vivo* decrease in gradient between the slow-release compound and its environment.

While initial difficulties were encountered with creating a sustained and visible flow through the vacuum, this was solved by the increased negative pressure in the vacuum tube (by removing 10 ml of air from a closed 3 ml tube). Enough volume of sample was obtained at each data point to obtain a concentration of the sampled fluid, even though a loss of negative pressure would have occurred at the time of switching out the tubes. The total amount of fluid was not recorded and no calculations for total amount of Ag removed were performed. The current study was designed with a need to obtain a large enough volume within 5 min, as opposed to continued recovery over a longer timeframe. In an earlier study in dogs with successful usage of UF probes (RUF3-12 similar to this study), an ISF sample was collected every 2 h (*Messenger, Wofford & Papich, 2016*), according to a schedule that increased in time (after 1, after 2, and after 4 h and more) (*Bidgood & Papich, 2002*; *Papich, Davis & Floerchinger, 2010*). One recent less successful *in vivo* study used a 24-h sampling timeframe for 21 days (*Partyka et al., 2024*), or an initially more frequent sampling timeline (*Munn & Whittem, 2024*) over 72 h. Several probe and sampling related complications and issues were reported in these two recent studies (*Partyka et al., 2024*; *Munn & Whittem, 2024*). Probe malfunction secondary to animal interference was reported in two rabbits (*Partyka et al., 2024*) but not in any of the eight sheep (*Munn & Whittem, 2024*), which is similar to the earlier canine studies (*Bidgood & Papich, 2002*; *Papich, Davis & Floerchinger, 2010*; *Maaland, Guardabassi & Papich, 2014*; *Messenger, Wofford & Papich, 2016*) and what was reported in pigs (*Messenger, Papich & Blikschlager, 2012*). Sample volume was reported as variable (sometimes insufficient for analysis) (*Partyka et al., 2024*) and was reported as poor in the second study (*Munn & Whittem, 2024*) with only one out of eight probes consistently producing a volume of more than 0.1 ml. These issues were not noted in earlier canine (*Papich, Davis & Floerchinger, 2010*; *Maaland, Guardabassi & Papich, 2014*; *Messenger, Wofford & Papich, 2016*) or porcine (*Messenger, Papich & Blikschlager, 2012*) studies. The skin entry points for the probes were not sealed in the sheep, with air leakage/loss of negative pressure hypothesized to be the cause for the low volume. A 4 ml tube was used in the sheep (*Robotti et al., 2020*), and the volume of the tube was not reported in the rabbit study (*Fulgenzi & Ferrero, 2019*), pig study (*Messenger, Papich & Blikschlager, 2012*) or the dog studies (*Papich, Davis & Floerchinger, 2010*; *Maaland, Guardabassi & Papich, 2014*; *Messenger, Wofford & Papich, 2016*). Similarly, no details were reported regarding measures performed to avoid air leakage in the earlier studies (*Papich, Davis & Floerchinger, 2010*; *Maaland, Guardabassi & Papich, 2014*; *Messenger, Wofford & Papich, 2016*; *Messenger, Papich & Blikschlager, 2012*). Direct trauma and loss of negative pressure appear to be the most reported issues with *in vivo* usage. For our *in vitro* study, allowing for longer sampling times might have negated the need for additional negative pressure, and the need for additional sealing. However, this might have increased the risk for different concentrations over time, theorized to occur secondarily to non-agitation of the fluid.

The concentration of Ag measured in the samples obtained both by UF probe and pipette sampling decreased over time. This could be explained by the stagnant nature of the fluids within the tube, although care was taken to sample from the lowest area of the fluid in case of sedimentation. Elution experiments ideally are performed with constant stirring (*Messenger, Papich & Blikschlager, 2012*), or agitation prior to sampling (*Bates et al., 2024*). We opted to not agitate the tubes in order to keep the probes fully submerged at all times. In addition, the probe would remain in the same place *in vivo* as well, and a stationary fluid was felt to be appropriate given that no concentration gradient was present. Pipette samples were obtained from the area in the tube in the location of filtration membranes of the UF probe as to not introduce a variable between the two methods. The higher variation seen in samples obtained by UF probes, especially in DPBS, is interesting and indicates that care must be taken with *in vivo* sampling to account for variability, for example by taking multiple samples at each time point. In prior *in vivo* studies, similar findings were observed. For example, the highest measurements ($C_{max}$) of florfenicol in ISF in rabbits (within 4 days) ranged from 54.72 to 638.23 μg/mL, or 245 ± 223 μg/mL (*Partyka et al., 2024*). For cefpodoxime measured in ISF in dogs the $C_{max}$ was reported as 4.33 ± 1.96 μg/mL (*Papich, Davis & Floerchinger, 2010*). The $C_{max}$ for meropenem in ISF in dogs was 24.32 ± 8.93 μg/mL after intravenous (IV) delivery and 10.95 ± 0.99 μg/mL after subcutaneous (SC) delivery (*Bidgood & Papich, 2002*). Interestingly, both the pipette and UF probe samples in a similar experiment with carboplatin solution in DPBS or plasma had variation, with no appreciable larger variation for the UF probes compared to pipette obtained samples (*Risselada & McCain, 2023*). While the relevance of this finding is unknown, there was a consistently reported range and variation in measurements in samples obtained *via* UF probes.

When expressed as RSD, the UF probe in DPBS underperformed compared to pipette sampling (195% *vs* 105%). The UF probe in plasma performed better (less variation), but still slightly worse than the pipette-obtained samples with an RSD of 54% *vs* 45%. Calculation of RSD of prior *in vivo* usage of UF probes amount to 91% for florfenicol in ISF in rabbits (*Partyka et al., 2024*), 45% for cefpodoxime in ISF in dogs (*Papich, Davis & Floerchinger, 2010*), 36% (IV) and 9% (SC) for meropenem in ISF in dogs (*Bidgood & Papich, 2002*), revealing a wide variation between studies.

Additional limitations could be inconsistency in mixing the specimens and solutions as well as sampling or measurement errors. We included three repeats of the elution study to minimize the effect of both inconsistency and sampling. Six repeat samples were used in the static sampling study, however inconsistency between the solutions (DPBS *vs* plasma) could still exist, although the direct comparison between sampling methods was based on the same solution.

## CONCLUSIONS

Elution into plasma and UF probe sampling of Ag in plasma are possible. While use of DPBS might be more cost effective, plasma should be considered for *in vitro* silver elution studies due to difference in elution and recovery. Silver nanoparticles in poloxamer 407 might release more slowly than other drugs that have been studied under similar

conditions. Both UF probe sampling and direct pipette sampling underestimated the anticipated Ag concentration in the solution. Ultrafiltration probe sampling of Ag is possible and UF probes could be used to measure *in vivo* local tissue concentrations, but the possibility of variations in the sampled fluid should be kept in mind. Increased number of samples or enrolled subjects and increased number sampling time points should be considered to decrease the effect of this potential variation. Similarly, the existence of variation between samples should be kept in mind, and UF probe obtained samples might be best used to observe trends over time rather than be relied upon for the absolute value of a single specific sample.

## ACKNOWLEDGEMENTS

The authors thank the Nanomedicines Characterization Core Facility (NCore) at the Center for Nanotechnology in Drug Delivery (CNDD) and Dr Marina Sokolsky for their assistance with sample and data analysis.

### Funding

The Purdue University Libraries Open Access Publishing Fund partially funded the APC. The funders had no role in study design, data collection and analysis, decision to publish, or preparation of the manuscript.

### Grant Disclosures

The following grant information was disclosed by the authors:
The Purdue University Libraries Open Access Publishing Fund.

### Competing Interests

The authors declare that they have no competing interests.

### Author Contributions

- Marije Risselada conceived and designed the experiments, performed the experiments, analyzed the data, prepared figures and/or tables, authored or reviewed drafts of the article, and approved the final draft.
- Robyn R. McCain performed the experiments, analyzed the data, authored or reviewed drafts of the article, and approved the final draft.
- Miriam G. Bates performed the experiments, authored or reviewed drafts of the article, and approved the final draft.
- Makensie L. Anderson performed the experiments, authored or reviewed drafts of the article, and approved the final draft.

### Data Availability

The raw data for this study is available in the Supplemental File.

## Supplemental Information

Supplemental information for this article can be found online at http://dx.doi.org/10.7717/peerj.18388#supplemental-information.

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
