# Peer review of "Silver nanoparticles can be sampled by ultrafiltration probe but elution into & recovery from plasma and Dulbecco's Phosphate Buffered Saline differs *in vitro"

_PeerJ, doi:10.7717/peerj.18388_

## Round 0.1 · original submission · Major Revisions

I am concerned with the comments orf Reviewer#2. However, I think the authors can respond properly that issue highlighted by Reviewer#2. Revise the structure to the manuscript, especialy the Abstract.

Reviewer 1 ·

Basic reporting

In their manuscript, the authors have shown that the silver nanoparticles can be eluted and probed with ultrafiltration (UF) into plasma. The language structure of this manuscript is decent, and previous works are properly cited as references. The authors have also provided their raw data, which is very helpful for readers to evaluate the experimental results. However, the structure of this manuscript is not very clear, and the presentation of the figures and tables is somehow ambiguous, which should be revised before the manuscript can be pulished. My suggested improvements are as follows:
[1] The author starts the abstract by directly discussing the experimental setup and results. However, I would recommend to remove those contents to later parts, for example, the materials and methods part or results part. The abstract should be a succinct summary of this work and the aim/goal of the project. Therefore, the authors should avoid adding too many experiment details to the abstract.
[2] The presentation of Table 1 is not very clear. For example, what does each row means? Why there are three numbers in each column? More details should be added to make the table easier to read.
[3] The presentation of Figure 1 is ambiguous. There seems to be 2 set of data in each graph, but no legends are added to the graphs, making it hard to understand what does each line represent. Also, some of the error bars are invisible, especially in graph A and B. I would recommend the authors to add the relevant information.

Experimental design

The authors have provided sufficient details for readers to understand their experiment processes and results, which is very good. However, I would suggest the authors to add several some schematics of their experiment to make things clearer. For example, they can add a flowchart as to show how the experiment is conducted and what each step is. Also, they are provide a schematic for their experiment setup.

Validity of the findings

Throughout their experiment, the authors have found that the silver nanoparticles can be sampled by UF probe and elution into plasma, which is meaningful. Also, the results the author have reported will be beneficial to later studies about the nanoparticles, as people can utilize similar methods to elute the nanoparticles in plasma and probe them based on this study. The authors have provided sufficient data to support their findings, which statistically agrees with the conclusion they have claimed. However, I would recommend the authors to add more discussions in the conclusions to show how their research can help the literature and future studies on similar fields.

Reviewer 2 ·

Basic reporting

Risselada et. al. have investigated the use of canine plasma for in vitro release studies of Silver Nanoparticles (AgNPs) from a suitable drug carrier construct such as poloxamer 407. They have contrasted this with Dulbecco’s Phosphate Buffered Saline (DPBS) which is conventionally used for such a purpose. Additionally, they have investigated the use of an Ultrafiltration (UF) probe as a sampling tool for each of these elution solvents. The authors hypothesize that the elution of AgNPs into DPBS will be similar to Canine Plasma and UF probe sampling will be similar to direct sampling. Overall, the article is clear, concise and unambiguous with professional language. Sufficient background information and contains appropriate references where necessary. Original figures, tables and calculated data were provided. Raw data from ICP-MS used for quantification was not provided. In my opinion, this is necessary to support the conclusions drawn from the data provided. The publication is self-contained with the relevant information to this article provided within. I appreciate that the authors have presented the materials and methods in great detail, and it would be easy for someone who wishes to replicate this study to do so, with the exception of the analytical methods.

Experimental design

As a part of their experiment, the authors encapsulated AgNPs in Poloxamer 407, a thermoresponsive polymer that undergoes sol-gel transition through micelle formation when temperature is increased. At the concentration used by the authors (20% post dilution) the polymer is a gel at 37°C where the experiment was carried out. The transferred the AgNP-Poloxamer 407 solution to a dialysis tube and eluted the AgNPs into either DPBS or Plasma. They sampled the solutions at multiple timepoints up to 96 hours using micropipettes and analyzed the silver content using ICP-MS. In parallel, they prepared two samples each with DPBS and Canine Plasma and sampled at multiple timepoints upto 1 hour.
Overall, the methodology and the rationale behind certain decisions is quite well explained. The content is within the scope of the journal. However there are some major fundamental concerns I have regarding the study.
1. The authors started the manuscripts with two distinct questions that are not related to each other in terms of the underlying process. The first hypothesis was that the silver nanoparticles elute into plasma same as DPBS. This phenomenon is governed by the transport properties of poloxamer gel and dialysis membrane to allow elution of AgNPs in different media. This is a release kinetics problem. The second hypothesis is that UF probes can be used to draw samples from either DPBS or plasma. This phenomenon is governed by the transport properties of the UF membrane and is unrelated to the release kinetics of AgNPs from the poloxamer construct. This is a sampling methodology problem. As such, these two questions, are well defined and motivated on their own. However, they are also fundamentally different. This is also evident from the difference in the choice of experimental setup authors chose to evaluate either problem – i.e. first question is investigated with poloxamer construct and direct sampling only; second question is investigated with a predefined stock solution and comparing between a UF probe and direct sampling. Therefore, they should not be combined into one manuscript.
2. Authors state that plasma has not been evaluated for Ag release studies (Line 43) and potential effect of EDTA in the analysis (lines 178-185). Therefore, the manuscript would benefit from raw data from atleast one of the quantification workflows in the supplementary material to demonstrate how the authors navigated these challenges during the analysis. It is also unclear if the authors compared quantification of AgNPs in plasma before and after nitric acid digestion to validate the quantification method.
3. In the UF probe sampling experiment (Table 1), the calculated starting concentration of the solution was 1247 ppb. Yet, in the control experiment of direct sampling the DPBS solution, none of the values are close to the starting concentration. Authors acknowledge this phenomenon (Line 155-157) and explain it by saying that they chose not to mix the tubes to make it representative in vivo sampling (Lines 225-230). Although discussion of limitations is appreciated, I find it to be inadequate by itself. This result does not inspire confidence that the quantification method was accurately able to measure concentration of AgNPs accurately. Since it was used throughout the paper and is an important pillar that supports the conclusions of this paper, additional experimental evidence validating the quantification methods must be provided.

Validity of the findings

Overall, without sufficient evidence that the detection method used is suitable it is difficult to conclude that the findings are valid or are supported by the results provided. Authors acknowledge this shortcoming multiple times throughout the article.

·

Basic reporting

The article presents a detailed study comparing the influence of different elution fluids on the release of silver nanoparticles (AgNP) and the efficacy of ultrafiltration (UF) probes versus direct sampling. However, to enhance clarity and robustness, it would benefit from addressing some redundancies in the result presentation and ensuring consistency in describing mechanism details, particularly since the releasing mechanism is not described within the study. Although the study showed that different elution fluids affect the release of AgNPs, however, it is also important to describe the reason behind the phenomenon, considering the aim of the study is to understand these influences. By addressing these anomalies, the manuscript can present a more coherent, non-redundant, and analytically rigorous discussion of the study’s significant findings and implications regarding the sampling approach and the influence of mediums.

Experimental design

1. The experimental design is robust, with clearly defined objectives and methods. The study investigates two main hypotheses: the influence of elution fluid on AgNP release and the comparison of UF probe sampling with direct sampling.
2. Use of both DPBS and canine plasma to simulate different elution environments.
3. Detailed description of the preparation and sampling procedures.
4. Appropriate use of ICP-MS for silver analysis.
5. The study relies on a limited number of specimens, which may affect the generalizability of the findings.
6. The choice of using EDTA plasma, despite its potential interaction with silver, could influence the results and was noted as a limitation.

Validity of the findings

The findings are presented with clarity and supported by the data. The study successfully demonstrates differences in AgNP elution between DPBS and plasma and evaluates the feasibility of UF probe sampling. The data is presented in a clear and logical manner with appropriate use of graphs and tables. However, there are some limitations, which the study should mention, for instance, the study notes that the UF probes had issues with initial negative suction, potentially affecting the reliability of the sampling method. And that the variation in Ag measurements obtained by UF probes suggests a need for further investigation to optimize this method.

Additional comments

1. The abstract contains long, complex sentences that are densely packed with information, making it hard to follow. Break down complex sentences into shorter, more digestible ones. Use bullet points or numbered lists where appropriate to clarify distinct points. For instance, line 20 “Six pipette and UF probe samples were taken of a 0.001mg AgNP/ml DPBS or plasma solution” is difficult to understand. And “s” is missing in the “six pipettes”. Please separate and clarify like “0.001 mg/mL AgNP were prepared in DPBS or plasma solution, then six pipettes and UF probe were taken out of these samples.” The flow of information is not smooth, and it can be challenging to understand the main findings and their significance at a glance. There are many specific details (e.g., measurements, time points, equipment settings) crammed into the abstract, which can overwhelm the reader.
2. Although methods are described in detail, allowing for reproducibility, however, these steps seemed very complicated in text, these approaches should be presented in abstract images for showing the steps for enhanced readability.
3. The manuscript could benefit from a more thorough literature review to include recent studies.
4. Some sections, such as the introduction, could be more detailed to improve readability. For instance, a brief introduction about the ultrafiltration probes, and the mechanism of such probes is necessary for better readability.
5. The introduction could be streamlined to improve flow. For instance, combining related information on AgNP and poloxamer 407 would make it more concise.
6. In the methodology part, it is necessary to explain the choice of a 1:2 ratio for AgNP is useful, but the manuscript would benefit from more detail on why this ratio was chosen over others based on preliminary results.
7. Considering the application of AgNP is for antimicrobial applications, the antimicrobial ability should be assessed by bacterial assays, as well as the biocompatibility by cytotoxicity assays.
8. The results section is comprehensive but could be enhanced by providing more context or comparison with previous studies.
9. The discussion is thorough but could include more on the potential implications of the findings for future research and practical applications.
10. The conclusion effectively summarizes the findings, but a stronger emphasis on the practical applications and future directions would be beneficial.
Overall, the study provides valuable insights into AgNP elution and sampling methods but would benefit from addressing the noted weaknesses to strengthen the findings and their implications.

---

## Round 0.2 · accepted · Accept

The authors have properly addressed all Reviewers comments. The article is ready for publication.

Reviewer 1 ·

Basic reporting

Compared to their last version of submission, the presentation of this work is much clearer. The literatures are now well cited and the backgournd information is thoroughly discussed. The article structure and figure presentations have improved, and the main conclusion is supported by the experimental results. The important raw data are also provided.

Experimental design

The disscussion of experiment is much more clearer than the last version of manuscript. I have no more comments to add.

Validity of the findings

The authors have provided sufficient data to suppot their conclusion, which is well validated by their experiment. The presentation of their data has significantly improved, and the figures/tables are much clearer. The conclusion is well linked to their main research question.

Additional comments

The revision of the manuscript is thorough and complete, I would now recommend the publication of this work.

Reviewer 2 ·

Basic reporting

The authors have addressed all the concerns I raised

Experimental design

The authors sufficiently address my concerns in the comments

Validity of the findings

The authors have addressed all the concerns I raised by providing the requested data

·

Basic reporting

Solved

Experimental design

Solved

Validity of the findings

Fine by now

Additional comments

Solved